# Divine Discomfort: A Relational Encounter with Multi-Generational and Multi-Layered Trauma

## Christoffel H. Thesnaar

Practical Theology & Missiology, Stellenbosch University, Stellenbosch 7602, South Africa; cht@sun.ac.za

**Abstract:** The COVID-19 pandemic has been, and will continue to be, exposed to young democracies who are grappling with deep-seated multi-generational and multi-layered traumas which are embedded in past and present conflicts as well as injustices. In the lead-up to the COVID-19 pandemic, South Africa, being a young democracy, has experienced an increase in anger, violence and vengeance due to on-going poverty, lack of service delivery, housing and land, etc., at all levels of society. The COVID-19 pandemic has exacerbated the injustices that are based on the legacy of generational trauma and pain that are caused mainly by an unjust political system, centuries of colonialism, violence, and conflict. These injustices have also exposed the commitment and promises that the religious sector made, during the hearing of the faith communities at the Truth and Reconciliation Commission (TRC), to the reconstruction and development of the country as well as to reconciliation and healing of the nation. Since South Africa became a democracy, the transition process, the first democratic election, the TRC process, political promises, corruption and currently the COVID-19 pandemic have restrained the state of trauma in the country. This restraint has led to a state of frozenness. This contribution argues that the concept of divine discomfort and specifically the notions of accountability and justice can contribute to exploring new ways for religion to deal with the eruption of multi-generational and multi-layered 'frozen trauma'.

**Keywords:** divine discomfort; multi-generational and multi-layered traumas; frozen trauma; violence; conflict; dialogical intergenerational pastoral process; responsibility; accountability; justice; COVID-19 pandemic; pastoral care



## 1. Introduction

In the lead-up to the COVID-19 pandemic, South Africa, being a young democracy (just more than 25 years), experienced an increase in anger, violence and vengeance due to ongoing poverty, lack of service delivery, housing and land at all levels of society. The temporality of almost 30 years after the transition in South Africa alerts us to an honest assessment of the impact of past traumas on the current population of South Africa. The COVID-19 pandemic has exacerbated those injustices that are based on the legacy of generational trauma and pain that are mainly caused by an unjust political system, centuries of colonialism, violence, and conflict.

These injustices have also exposed the commitment and promises that the religious sector made, during the hearing of the faith communities at the Truth and Reconciliation Commission, to the reconstruction and development of the country as well as to reconciliation and healing of the nation,. Since South Africa became a democracy, the transition process, the first democratic election, the TRC process, political promises, corruption and currently the COVID-19 pandemic have reiterated the effect of past trauma in the country. During a Gala Dinner hosted by the Institute for Healing of Memories at the end of 2018, Mamphela Ramphele[1] (Ramphele 2018) indicated that our democracy remains on shaky ground. According to her, South Africans have failed dismally to complement our ground-breaking political settlement of 1994 with what she calls an emotional and socio-economic settlement. With the phrase 'emotional settlement', she implies the healing of the

traumas of apartheid and the embrace of the values of ubuntu, while with 'socio-economic settlement'[2], she implies the dismantling of structural barriers to equality in our society.

It is safe to say that in the process of the development of the young democracy in South Africa over the last 30 years, the current impact of the COVID-19 pandemic has contributed to suppress the state of past trauma in South Africa. This has literally kept the past trauma in what I want to call a 'state of frozenness'. The lack of urgency among all roles in transforming South Africa at all levels has led to a painfully slow process of transformation and an alarming increase in divisions between rich and poor, different race groups, leadership and the people, etc. There is a profound danger related to frozen trauma[3], as we know that the failure to deal with past trauma can lead to relational destructive behavior or to an eruption of suppressed anger, violence and vengeance at any moment.

The current state president, Mr. Cyril Ramaphosa, at the funeral of Miss Winnie Madikizela-Mandela in 2019, acknowledged the failure in dealing with the traumas of South Africa's past (Ramaphosa 2018). He indicated that South Africa needs a new language to speak about its traumatic past and that failing to attend to it will continue to impact current and future generations. He used words such as 'hurt', 'pain', 'woundedness' and 'anger' to voice the trauma experienced by the majority of the people in South Africa and emphasized social healing. With this new language, he acknowledged the legacy of generational trauma and pain in society, mainly caused by an unjust political system and centuries of colonialism, as well as how this inheritance is transmitted to the current generation and will continue to be transmitted to future generations if it is not transformed.

The intense reaction to the 2019 publication of four studies from Stellenbosch University that purported to demonstrate that 'coloured' women have lower cognitive development than the rest of humanity, as well as the fact that the study was even attempted, again confirmed the impact of the trauma of apartheid on South African society. In her reflection on this tragic publication, Mamphela Ramphele affirmed that woundedness of South African society was again laid bare by the publication of this study and therefore the task of healing this nation is urgent. She ended her reflection by stating: "The long-term future of our country will be shaped by the extent to which healing our wounds and discarding colour coding is tackled as a priority to free the human potential of all citizens" (Ramphele 2019). This restraint confirmed the "state of frozenness" in which individuals, communities and the nation currently find themselves. In addition, the initial impact of the current COVID-19 pandemic has exacerbated the past injustices in South Africa and therefore further contributes to the state of frozenness of past trauma (Thesnaar 2021, p. 97). This contribution aims to determine whether the concept of divine discomfort (coined by philosopher Levinas) and specifically the notions of responsibility, accountability and justice can contribute to exploring new ways for religion to deal with the eruption of multi-generational and multi-layered 'frozen trauma'.

Based on the argument thus far, this contribution seeks to determine whether the concept of divine discomfort, supported by the theory of the Dialogical Intergenerational Pastoral Process (DIPP), particularly the notions of accountability and justice, can contribute to the search for ways to deal with the eruption of 'frozen trauma'. Although it is beyond the scope of this address to describe the DIPP in detail, it is necessary to state that it is based on the contextual theory developed by the Hungarian psychiatrist Boszormenyi-Nagy, the practical theological and pastoral care theory of Meulink-Korf and Van Rhijn as well as the theologies of Mugumbi, Gathogo and Mkhize.[4]

## 2. Impact of Past Traumas

We know that neuroscience has now established beyond a doubt that the consequences of unattended trauma can manifest in impulsive, abusive and violent behaviour (Chakraborty et al. 2007). Of all the forms of trauma, humiliation[5] has the greatest impact on a person's psyche. This is particularly the case in authoritarian patriarchal societies where hierarchical structures at home, in communities and at educational and workplaces

reinforce experiences of retraumatisation. According to Mamphela Ramphele (Ramphele 2018), South Africans are reaping the bitter fruits of the colonial apartheid migrant labour system and job reservation, which placed black people at the bottom of the pyramid of a white male-dominated system. Poor men repeatedly humiliated by not being able to provide for, command and control their households might run amok and commit unspeakable crimes, or be perpetuating violators (Ramphele 2018). Therefore, where poverty prevails in the midst of the wealth of a few it will always tend to lead to further wounding of individuals, families, communities and a nation at large. During the time of hard lockdown due to the pandemic, families and communities were particularly exposed to domestic and partner violence, bearing witness to the eruption of the deep-seated trauma of many individuals and communities with devastating effects (Dartnall et al. 2020, pp. 1–2). The devastating increase of domestic violence during the pandemic is a global phenomenon and therefore not limited to South Africa (Halloway 2021; Shandilya 2021).

This is why domestic and intimate partner violence is so brutal[6], why violent crime so common, why substance dependency has become a lifestyle, why xenophobia is a devastating reality, and why gang violence is a fearful alternative to humiliation within societies.

A further indication, post transition, is the emergence of the next generation of South Africans, a born-free generation, if you wish, that is not afraid to challenge the lack of transformation at all levels of South African society. They are not afraid to challenge the demographic transition[7], the TRC process, the failure of attending to the recommendations of the TRC by all sectors of the society, the role Mandela played during the transition, the failure to establish a socio-economic settlement from which all South Africans can benefit, the elders of the struggle, corruption and government on neglecting the constitution of the country. One of the ways this born-free generation came to the fore was in the 'fallist movement'[8] that played a major role in violently challenging colonialization and demanding the decolonialisation of institutions. This fallist generation as well as the impact of the COVID-19 pandemic has alerted South Africans to the fact that it is continuously paying a high price for the failure to invest in healing its multi-generational, multi-layered, wounded society.

The gravest danger of the impact of frozen conflict is not that it erupts in the form of an increase in hostility, violence, conflict and abuse between human beings, families, communities and countries as this is witnessed in the public domain. The gravest danger that the effect of frozen conflict has on an individual, family or group is that it is suppressed, hidden, and that the individual, family or group do not realize the impact it has on their relations with themselves and the other. In many ways, they tend to deal with the impact of the frozen trauma to create an exclusive life for themselves without the other, where they only see the other as objects in their world (I–it relations, according to Buber (Buber 2004). What this attitude boils down to is that an individual, family or group can be so arrogant that they tend to live their life only according to their own needs and therefore indicate that they do not need the other. For example, I/we think I/we have a right to more space, recognition and authority; I/we think I/we am entitled to what I/we need without the other; I/we strive at all costs to be recognized, at the expense of the worth of the other. As Boszormenyi-Nagy and Krasner (1986) emphasize, "people use each other, are used by each other, and accept or fight against particular usages of each other". In a sense, to state it bluntly, humans have the inclination to destroy the other at all costs, as we strive to be better than the other is and are not just happy to be equal to the other.

Van Riessen (2018) refers to the work of Krznaric, who confirms that our complex brains are wired for both individualism and empathy, but that the individualistic side has been too central in the past three centuries. This is based on a particular assumption that life develops and grows from the basic expectations you or I have. In other words, we measure the value or quality of relationships in accordance with the expectations we have and the comfort with which we want to live. Our homes become a comfortable fortification where we live and meet with only those with whom we want to engage. Our homes

and our relationships are defined by our own beliefs, language and ideas. We create a separate world that is as comfortable as possible away from the rest of the world we live in. The implication of striving to live in comfort is that it blinds us to the discomfort of the other. The more comfortable we become, the more our homes become the basis for exclusion, economic status, classism, racism, religious prejudice, division, moralism, etc. It is in our homes that the relational gap between us and those whom we see as objects widens, something that has continuous implications for the way we live life outside our homes. These individualistic and group isolations and privatizations are also an indication of frozen trauma that has started to erupt. Therefore, the more comfortable individuals and groups in South African become, the more they are unable to hear the discomfort of the other and themselves, and the more they are capable to close their ears to the discomfort of the other.

Based on the argument thus far, South Africans are currently reaping the fruits of frozen trauma that has started to erupt in society and the country. Although South Africa went through a transformation (political change and a new constitution) and healing process (facilitated by the TRC) there was no guarantee that it would be sufficient to deal with the centuries of frozen trauma. The lack of implementation of the TRC recommendations by government, civil society and religious groupings, the failure to facilitate the past trauma and the failure to address socio-economic settlement in terms of economic justice, land reform, housing and employment, to name a few, has specifically contributed to the frozen trauma and the subsequent eruption thereof.

## 3. Divine Discomfort

Jewish philosopher Emmanuel Levinas[9], a religious scholar with a strong focus on love for the widow, orphan, stranger or neighbour, focused his work on the encounter with the other. To Levinas (1997), "the good assigns the subject, according to an assumption that cannot be assumed, to approach the other, the neighbour. This is an assignation to a non-erotic proximity, to a desire of the non-desirable, to a desire of the stranger in the neighbour" (Levinas 1981). The philosophy of Levinas could be described as ethics. However, as Bergo (2011) correctly states, "[i]f ethics means rationalist self-legislation and freedom (deontology), the calculation of happiness (utilitarianism), or the cultivation of virtues (virtue ethics), then Levinas's philosophy is not an ethics". The ethics of Levinas is rather the development of a 'first philosophy'. Bergo (2011) describes this as follows, "It is an interpretive, phenomenological description of the rise and repetition of the face-to-face encounter, or the intersubjective relation at its precognitive core; viz. being called by another and responding to that other."

In his ethics, Levinas wrote eloquently about the face of the other. The face is an epiphany (holiness). Levinas indicated to Derrida in a conversation: "You know, one often speaks of ethics to describe what I do, but what really interests me in the end is not ethics, not ethics alone, but holy, the holiness of the holy." (Caruana 2006). The holiness of the other is holier than the land, even if that land is the Holy Land. He deliberated whether it should be a face or a mask of the other. To him the face of the other cannot be a mask. The face of the other is not something you can draw. It is an appearance. It is only when we are able to see the face of the other, that we will be transformed by its vulnerability so that it becomes almost impossible to ignore the discomfort of the trauma of the other on us.

For Levinas this is ethical and because it is ethical, we as humans should always find a way of engaging with the other. According to him, the encounter with the other will help us to abandon our ego project and to live with compassion for the other. What does this entail? To Levinas (1978), "[t]he other as other is not only an alter ego; the other is what I myself am not". With this statement, it is clear that Levinas does not see the other as an alter ego, a replica of the self. The key for him was to let the other be other as other. He wrote: "If one could possess, grasp, and know the other, it would not be other" (Levinas 1978). The face of the other resists the fact that I might be able to possess or grasp it. It also resists my power to control or manipulate it. Levinas indicated that the relationship with

the other is always asymmetrical; therefore, he referred to the height of the other. To him the other and their needs always stand above me. We are never equal. The other is what commands me. He says the house that is closed to the other is not ethical. It is therefore only by others that I am free and liberated from my ego self. The shortest road to me is through the other.

It is clear that for Levinas, 'the other' is not 'just' a neighbour. The other is foreign, we can also say strange to me in many ways. This foreignness and strangeness is not a regrettable deficiency. On the contrary, this foreignness, strangeness and otherness provokes and stimulates responsibility (Klun 2011). Van Riessen (2018) indicates as follows: "[In this] phenomenology of responsibility, Levinas holds that the ethical relation between self and other is inspired by the absence of the possibility of fusion. It is the 'alterity' of the other, the strangeness of his or her appearing in an otherwise familiar and orderly world that commands the 'self,' the 'I' to take responsibility for the other." In this, regard it is essential to emphasize that the irreducible otherness of the human person does not entail the inexistence, but rather presupposes a common, shared humanity. The same, complete humanity is embodied in each individual in a unique and irreplaceable way. This unresolved and unresolvable tension between common humanity and singular individuality forms the necessary precondition to life-giving relationships between persons understood as infinite mysteries.

Levinas' ethics of responsibility makes an appeal to 'me' as an individual but also to 'us' as a collective. We are actually involuntarily chosen and even elected to do what is necessary to help the other (Van Riessen 2018). It is this overwhelming responsibility towards the other that Levinas (1981) describes as 'divine discomfort'. His emphasis on responsibility is because we are in a sense called into responsibility by the other (Caruana 2006). In this sense, I concur with Kosky (2001), who indicates that this responsibility would be the image of God in man, which is why Levinas calls this 'divine discomfort'. Responsibility entails that we always see the other as a human being. The face of the other speaks to you without even realizing it. Even the voiceless face speaks to you. It is something that comes to you that you did not even expect. Being human is therefore to be addressed and to be called, and to answer Hineni[10], Here I am. In this sense, we do not have a backpack filled with humanity. No, we are called into humanity. I need to answer the call, therefore I am (respondeo ergo sum). We need to hear that there are more others calling us. We are not alone or alone with one other person in the world. In their book The Unexpected Third, Dutch theologians Meulink-Korf and Van Rhijn (2016) add a further aspect to divine discomfort by calling it good faith (in other words chèsèd)[11] and compassion for the other.

What does it mean to open one's ears in order to hear the other and to take responsibility for the other in the context in which we live today? Does this mean that we need to be more hermeneutical in terms of trying our best to understand the other? Does this mean that we need to be focused on the needs of the other, the neighbour, and to make sure we recognize their needs and develop a plan of action? Does the focus on the other mean that we now need to be close to the other in order to have compassion, hospitality, empathy and interpathy with the other? It is of the essence that one will need to respond with great caution to these questions, as being close to someone or even in their proximity does not automatically mean that one is hospitable, compassionate or empathic. Van Riessen (2018) indicates: "On the contrary, Levinas seems to argue that experiencing responsibility or ethical engagement precedes empathy." This is key in the way one needs to understand our role towards the other. It is not about understanding the other and the frozenness of their trauma or the reasons why it erupted in order to assist, to care or to show compassion. It is also not about the arrogance to claim that we know the other or the trauma of the other. It is also not only about our actions to address the trauma of the vulnerable, even if our actions are very sincere and we want to make a difference. It is also not about a hidden goal to unify or unite opposite sides or to find mutuality or shared feelings about trauma. Levinas indicates that something needs to take place in us before we even think or do. In

other words, a movement needs to take place before we have even started to do anything. That movement is grounded in ethics, as indicated earlier. To unpack this further, Van Riessen (2018) explains that it rather has its origins in a shared vulnerability, being exposed as beings of flesh and blood to the same condition. The movement that leads to a shared vulnerability with the other is not only when there is violence, conflict or suffering, but also when there is peace and joy. Van Riessen (2018) emphasizes that in proximity, the strangeness of the other person is maintained; it is not cancelled, removed or nullified by the nearness. Responsibility is therefore to live in the discomfort of the trauma of the other. In short, this is called divine discomfort.

## 4. Accountability

Responsibility and accountability are relational words and concepts. Both are a discomfort because they are not something with which we as humans always want to be afflicted. We will rather try our best to avoid responsibility for the traumas caused by past generations. We will even develop theology and use God to exempt us from this responsibility so that we can avoid accountability at all costs. The parable of the Good Samaritan is to some extent an example of struggling to constantly live with divine discomfort for the other. In other words, are we able to stay in the discomfort of the other and answer the calling? The Jews interpreted the one that is near to me as a fellow Jew—the same people as my religious community. The Pharisees went further to exclude the ordinary person from being my neighbour and the Qumran community went even further to exclude those who were termed 'sons of darkness'. In general, Jewish usage excluded Samaritans and foreigners from possibly being called their neighbour. We could reason that the priest was entitled to pass by for fear of defilement and its consequences, as he was supposedly on his way from or to perform religious rituals. Although it may have been true that the priest and the Levite had ritual obligations that prevented them from acts of love, just as in the case of Jesus' healings on the Sabbath, essentially the point is that they failed to demonstrate love to the stranger, the other, irrespective of what the pretext was. The Samaritan also had laws and rituals to which he had to abide; however, whatever his obligations were, the point is that he abided the law, showed compassion and did justice to help the man (Marshall 1978). One could reason that because the Samaritans knew what it was to constantly live in discomfort, they had eyes and ears for the trauma of the other. It is a futile exercise for us to spend hours to attempt to find theological reasons why we are not able to take responsibility for the other, as no matter what we do, we are called to and can only be a neighbour to others. The question we really need to ask is, Am I the neighbour for those who are in pain? How can I be a neighbour to the traumatized other? How can I stay in the discomfort of the trauma of the other, and show compassion and do justice?

It is in this sense that accountability calls us to order, as we cannot just do what we want. Accountability belongs to the domain of ethics; it is relational and therefore it is ultimately about a commitment to sustainable encounters with the other. The other helps me as a person to discover what Levinas calls the 'humanness' of the other person. For Levinas, humanness is not knowledge about God, but rather the place where God works, where 'God lives'. In this regard, we should understand compassion as an actual appearance (epiphany) of God's presence in the human being and therefore the human being is fully human and at the same time can rise above mere humanity (Van Riessen 2018). My humanness is therefore in my accountability. What does it mean to recognize the humanity of the other? As we are all recognized in the humanity of God, dignity becomes the value of the other. Then I/we will allow the other to show me/us just where my/our duty is. The appeal of the other to me/us presupposes that I/we are human, that I/we are accountable and that I/we will be open to listen and respond.

The recognition of humanness and dignity has strong African roots. When a Zulu[12] person greets another person with the word Sawubona, it does not simply mean 'hello' or 'I greet you'. It literally means 'I see you' or 'Until you see me, I actually do not exist'. When you/they therefore see me/us, you/we bring me/us into existence. I/we see the love and

your/their feelings and your/their soul and you/they mean everything to me/us. Pumla Gobodo-Madikizela (2018) also alludes to the value of using the expression Ndizakuthi ungowasemanini kanene? (To which clan shall I say you belong?). By asking this question and not merely asking what your surname is, you are actually recognizing the ancestral lineage by which the person can be identified as well as emphasizing the significance of the collective. In this way, the dignity of the person that is rooted within a line of honourable others and whose names are imbued with meaning are recognized. Mutual recognition inspired by *Ubuntu*[13] is fundamental to being a fellow human being, a relational subject in the context of community (Gobodo-Madikizela 2018). Embodied recognition of the mutual others will therefore actively seek to repair the brokenness of the other, as it has also become my/our own brokenness. It is essential to state that embodied recognition of the other's trauma does not involve appropriation, for this would imply the infliction of further violence and trauma, perpetuating and reinforcing oppression rather than healing and overcoming it. This is only possible if we are able to stand and sit in the other's brokenness. This will then transcend my own/our brokenness. This is a way to show respect in order for the stranger to have agency in the encounter. Gathogo (2008) argues that according to him the ubuntu philosophy would be described by the African concept of hospitality. Gathogo (2008, p. 285) indicates, "Ubuntu, however, requires an authentic respect for individual rights and values and an honest appreciation of diversities among the people."

Gobodo-Madikizela (2018, p. xxii) indicates that Ubuntu is defined by the multiplicity of relationships with others. Mkhize (2016) accentuates dialogue and relationship as to him, in relationships, I am not me, but what I am is community. My role is therefore to contribute to other people instead of only to myself. Dialogue does not start with me, but beyond me. It starts with my ancestors. I always need to ask, what are the expectations from them to the work I am doing with others? What was I told by them to do? Am I still in line with them? In this regard, hospitality and interconnectedness are key to dealing with the past and present traumas as well as possible future traumas. According to Mkhize (2016), consultations and rituals are of the essence for the healing of the unjust South African society. The TRC had the goal of contributing to individual and collective healing by exposing the truth and in this way assisting victims to get closure. However, the sad reality is, according to Mkhize, that the TRC did not invite the traditional leaders into this space to facilitate individual and collective healing in South Africa.

We need to acknowledge that it is always a challenge to be called, to be present and to be available to the other (kenosis). In Philippians 2:7 we read how Jesus answered the call to live in divine discomfort by being prepared to empty himself (literary becoming nothing, kenosis) from his will and becoming entirely receptive to God's divine will—to be a 'here I am' and to stand back and embody an 'after you' or 'you first' ethics, as indicated by Levinas. The self needs to become the host (ethics of welcome) and then the other as stranger automatically becomes the one whom I want to welcome to my home and country (Saldukaitytė 2019). Van Riessen (2018), however, adds that compassion should not exclusively be understood as a form of empathy or Einfühlung, or as a way to come near to the other through understanding. Instead, it should be seen as the ability to engage with others despite their strangeness and incomprehensibility. Therefore, once we encounter the face of the other, responsibility arises and that implies justice. In this regard, Meulink-Korf and Van Rhijn (2016) indicate that it does not matter who the other is when we have an eye for experiences of having received and experiences of giving (sensitivity to good faith and compassion from ourselves to another); we are already moved even before we decide to show compassion to the other ('passivity'). Compassion will then need to take the place of the trauma of the other generations and be patient[14] with the other generations, even if their trauma is radically different.

In the contextual theory, Boszormenyi-Nagy places emphasis on generations and indicates the important influence of three generations in the current and future reality in which people live. According to Boszormenyi-Nagy, the past, present and future are closely linked and, in that sense, tomorrow starts yesterday. The key is that we are all related to

the previous and future generations. Meulink-Korf and Van Rhijn (2016) remind us that in the relational reality of the here and now (the current entanglements between us), the past and future are not absent. It is therefore not unusual for the current generation to bear the responsibility of the actions and injustices of past generations. We should see ourselves as the in-between generation, the second generation. The Bible refers to the third and fourth generation and not to the second, because we are the addressed generation. As the addressed generation, we are responsible for the way we respond to the traumas caused by injustices in the past; continued injustices that cause trauma in the present and what we pass on to the next generation. We will also be held accountable if we choose not to respond to any of these traumas. In the same regard, we, as the current generation, also bear the responsibility for coming generations. This is a responsibility we did not ask for, neither did we agree to it; it is a real claim, whether we like it or not. Responsibility calls for taking a significant risk to engage with the previous and current generations regarding past traumas. This is possible, as transcendence implies that the drive for taking responsibility needs to be directed from the out-side, and not from within.

The claim does not fit into the frame of a contract between persons, groups or communities where the rights and obligations of both parties are clearly (mutually) described (Meulink-Korf and Van Rhijn 2016, p. 39). The claim in this regard is fundamentally based in ethics and in relations. Therefore, the claim is best described within the understanding of a covenant (Hebrew: berith). According to Meulink-Korf and Van Rhijn (2016), the basic words for human relationships (justice, reliability, truth, fidelity) are linked with a covenant and involve reciprocity with room for something extra, the free extra, the unearned, which knows no conclusive description or accounting. Mugambi (2012) affirms the importance of the covenant when he states that covenantal relationships are much more durable and sustainable than contractual ones. Within this context, Mugambi (2012) then emphasizes that the notions of forgiveness, repentance, gratitude and kindness are covenantal and not contractual. In this regard, the claim in a relationship is truly about accountability and therefore primarily ethical. Levinas is, however, very careful with the concept of relationship. The reason why he is so careful is that a relationship cannot be reduced to a trade deal of some sort, where I/we can stand in for you/them and then you/they stand in for me/us. My/our personal responsibility is already in place before there is any relationship.

## 5. Justice

I have argued thus far that our need to take responsibility and be accountable is rooted in ethics and in relationship with the other. Divine discomfort is about real dialogue with the other and between one another. We tend to describe dialogue as a meeting with the other based on love and therefore we need to acquire the ability to understand the other and to interpret the others' feelings in order to assist and care for them. However, this kind of understanding of love is actually a misrepresentation of the meaning of love and the meaning of love within the emphasis on relationship. Buber (Friedman 2008) helps us to understand that dialogue is not something you can organize; it happens by itself. When Buber indicates that there is healing through meeting, he emphasizes relationship as we are embedded in one another. Levinas's understanding of proximity is not the ability to know or understand the other, but rather as a difficult form of love, that can even endure the absence of equality and deep differences between the self and the other. In that sense, each meeting with the other is the light of God that shines in, and therefore it is about justice. So, we need to link justice with agape (love). I concur with the phrase that love also demands justice, and that my relationship to my neighbour, the other, cannot be different or separate from the relationships that my neighbour has to others. This phrase 'love demands justice' or stated differently 'love demands justice in action', is nothing new within the virtue tradition. Aristotle affirmed, many centuries ago, that the (cardinal) virtue of justice acts as a necessary precondition for healthy human interactions, loving relationships and happiness.

In this regard, justice indicates that I am also responsible for the responsibility of the other. The concept of justice alerts us not only to our responsibility to face and deal with the frozen trauma as well as the erupted trauma we currently experience; Boszormenyi-Nagy warns us that debt of the past traumas can be carried over from the one generation to the other and that this can have a destructive impact on current and future generations as they try to deal with the past debt. One person's deeds can hurt the collective family, society and world. In this regard, Martha Cabrera (2015), a Nicaraguan social psychologist, indicated the effect of trauma on the coming generations when she stated:

> "Trauma and pain afflict not just individuals. When they become widespread and ongoing, they affect entire communities and even the country as a whole . . . the implications are serious for people's health, the resilience of the country's social fabric, the success of development schemes and the hope of future generations."

Because of these deeds, we need to restore human justice. This is essentially about answering the calling of those crying for justice. In this regard, current South Africans, as the addressed generation, are not only responsible for dealing with the traumas of the past generations, but ultimately, they are also accountable to future generations and therefore the addressed generation owe it to them. If the South African nation continues to neglect this responsibility, they will carelessly pass our unresolved traumas on to the next generation. In this way, they will simply keep the frozen trauma intact and leave it for the next generation to deal with.

The demand for justice ('human justice') really starts when I/we become aware of my/our humanity and my/our natural injustice, the damage I/we inflict on others through my/our Ego structures (Meulink-Korf and Van Rhijn 2016, p. 94) At the same time, I/we also become aware of the injustices of previous generations for which I/we need to take responsibility and be accountable to. I/we cannot encounter with myself/ourselves and therefore I/we need to meet and dialogue with the current and future generations in terms of my/our responsibility towards justice to the other. In this regard, justice implies 'distributive' justice, which entails that South Africans cannot only limit those others to whom they want to address the multi-generational and multi-layered traumas, but they are also required to think about all the others with whom they are also connected. Moreover, it also concerns justice to the others for whom the other is responsible. Responsibility and accountability do not only entail distributive justice, but also transformative justice and restoration.

## 6. Conclusions

South Africans and other similar societies cannot afford to continue to ignore the unfinished business of the past and present traumas that continue to jeopardize the healing process of a nation. It is critical that countries facing similar challenges as South Africa, and faith-based communities in particular, should be vigilant for the signs and symptoms of multi-generational and multi-layered frozen trauma that is either still frozen or that has already erupted. The fact that societies are able to identify, remember, experience and see the past and present injustices should cause discomfort. However, faith-based communities and the leadership they have and can exercise in enabling the processing of traumatic experiences are strained through the ages. In many cases, these traditions and communities contributed to the infliction of trauma, through colonization and oppression. It is therefore key that these communities should not only be vigilant for the signs and symptoms of frozen trauma but that they in particular need to become aware, acknowledge and restore their own traumatic and traumatizing history before they are able to become agents of healing. Healing and transformation begin with oneself.

To address the impact of frozen trauma, faith-based communities need to create courageous spaces where individuals and collective groups can engage multi-generational and multi-layered frozen trauma with the other. To create these spaces faith-based societies need to embrace the values of Ubuntu and mutual recognition. They should avoid the trap of dealing with past traumas in an individualized way, as this will exclude the relational

engagement with the other. In this contribution, I have specifically argued that there is no place for individualism, as 'the other' in every society calls faith-based communities in particular to be accountable because they are related. Addressing the multi-generational and multi-layered frozen trauma calls for faith-based communities (individually as well as collectively) to face the past and current traumas, actively take responsibility for the injustices done and be accountable to the other, as this is key to the ethics indicated in the DIPP.

The DIPP approach starts from the assumption that human beings are meaningfully created in the image of God and therefore are bound to be meaningful for others and to give meaning to others. The uniqueness of Levinas is his emphasis on the Good, which arises in the midst of suffering and trauma as 'for the other'. As the addressed generation, living in discomfort, everyone should always be reminded they are not only part of the current society for each other or for themselves, but especially to the next generation(s).

Moreover, faith-based communities should realize that they are called to be accountable and just in their ministry to the other. As the addressed generation, they need not actively work towards the coming generation as people who struggle on their own, but because the transcended, God, is faithful, as stated in Isaiah 63:9: "In all their suffering he also suffered, and he personally rescued them. In his love and mercy, he redeemed them. He lifted them up and carried them through all the years." As the addressed generation, based on their responsibility and accountability to the past, to the current and to future generations, faith-based communities need to be continuously disturbed by the traumas of others and it should cause real discomfort in order to make it possible for the light to break through the frozen and erupted trauma. Where faith-based communities in their daily ministry succeed in being disturbed by the traumas of the other they are able to contribute toward breaking the cycle of intergenerational trauma. In this way, they will not only be able to actively contribute to dealing with the frozen trauma but also to actively contribute to sustainable peacebuilding. This entails uncovering our original ethical vulnerability: living in divine discomfort.

**Funding:** This research received no external funding.

**Conflicts of Interest:** The author declares no conflict of interest.

## Notes

[1] Mamphela Aletta Ramphele is a South African politician, an activist against apartheid, a medical doctor, an academic and businesswoman. She was a partner of anti-apartheid activist Steve Biko, with whom she had two children. She is a former Vice-Chancellor at the University of Cape Town and a one-time Managing Director at the World Bank.

[2] The violations of socio-economic rights were explicitly excluded from the TRC process.

[3] For a more indepth discription of this concept please see Thesnaar, (in Nel et al. 2020, p. 113).

[4] Ivan Boszormenyi-Nagy was a Hungarian psychiatrist who developed the contextual theory based on relational ethics. Hanneke Meulink-Korf and Aat van Rhijn translated the contextual theory within practical theological and in particular pastoral care. They particularly introduced the work of Martin Buber and Emmanuel Levinas to the contextual theory. Jesse Mugumbi and Julius Gathogo are two of Africa's leading theologians. Velaphi Mkhize is a poet, author and African healer/spiritualist. He is the president and founder of both the Umsamo African Institute and the South African Healers Association.

[5] See (Dlamini 2010). He indicates that black people were 'nothing more than objects of state policy'.

[6] We live in a world where we have neglected the key issues described in the preamble of the 1996 Constitution. "They are to heal the divisions of the past and establish a society based on democratic values, to lay a foundation for a democratic open society in which government is based on the will of the people and every citizen is equally protected by law, and to improve the quality of life of all citizens and free the potential of each person."

[7] The transition from a White minority political system to a Black majority political system.

[8] The fallist movement was a nation-wide movement that started in 2015 with the #FeesMustFall movement. This movement was the result of the growing dissatisfaction and disillusionment among young South Africans with the many empty promises of freedom, and a profound commitment to take up the challenges of this generation.

[9] Born 12 January 1906 in Kovno; Russian Jew; Lescourret referred to him as a "man of the borderlines". In 1923, he started his studies in Strasbourg. He became friends with Maurice Blanchot during this time. From 1928 to 1929 he went to Freiburg to

take courses of Husserl and Heidegger. In 1930, he became a French citizen. He wrote his first work, *On escape* (De I'evasion) in 1935. During the Second World War, he was a prisoner of war from 1940 to 1945. In 1947, he wrote *Existence and existents* and in 1948, he followed that with *Time and the other*. He was the director of L'Ecole Normale Israelite Orientale in 1947 and later became professor of Poitiers in 1967, the University of Nanterre in 1967 and then Sorbonne in 1973. He studied the Talmud and was influenced by writings of Dostoevsky. His most important works was *Totality and infinity: An essay on exteriority* (1961) and *Otherwise than being or beyond essence* (1974). His books were very difficult to read and interpret. After the war, he decided never to put his foot in Germany again and kept to his word until his death. He passed away on 25 December 1995.

[10]   *Hineni* is Hebrew for "here I am," and is the response Abraham gives when God calls on him to sacrifice his son Isaac.

[11]   *Chèsèd* is a Hebrew word. In its positive sense, the word is used for kindness or love between people, for piety of people towards God as well as for love or mercy of God towards humanity. It is frequently used in Psalms in the latter sense, where it is traditionally translated as 'lovingkindness' in English translations.

[12]   The Zulu people are an Nguni ethic group in South Africa and live mainly in the province of KwaZulu-Natal.

[13]   Xhosa: *umntu ngumntu ngabanye abantu* (which, loosely translated, means 'a person is a person through other people').

[14]   Levinas indicates that "[t]he modern world forgets the greatness of patience. The rapid and effective action, which put everything at risk at once, has made the hidden splendour fade of the ability to wait and to suffer. However, the glorious expansion of energy is murderous. One has to remind of the excellence of patience; not by preaching resignation against the revolutionary spirit, but in order to make one feel the essential bond that connects true revolution with the spirit of patience" (Levinas 1969).

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
