# Peer review of "Divine Discomfort: A Relational Encounter with Multi-Generational and Multi-Layered Trauma"

_religions, doi:10.3390/rel13030214_

Round 1
Reviewer 1 Report
The manuscript addresses the important topic of trauma processing at the level of entire cultural, political and religious communities and societies. It invokes South Africa’s post-TRC experience and history as context and source to propose a multi-layered understanding of trauma and its processing (or non-processing). Following Mamphela Ramphele’s lead, the author argues for the need to supplement political settlements (as a result of the TRC process) with “emotional” and “socio-economic” settlements intended to respectively address the healing of past traumatic experiences and the dismantling of structural/institutional biases, prejudices and discrimination. Using a multidisciplinary methodological approach, the manuscript proposes to respond to the experience and reality of “frozen trauma” by nurturing what the philosopher Emmanuel Lévinas has named “divine discomfort” through which robust notions of responsibility, accountability and justice are embraced and lived out. The invocation of contextual theology and African perspectives in particular, enables the adaptation of the notion of divine discomfort to the South African experience and reality. Such post-traumatic constructive theological reflection is especially relevant (and even urgent) at this time where a global pandemic draws most of the attention and focus of individuals and societies, relegating to the background other forms of (lingering) trauma (past and present) transmitted across generations without being properly addressed. The manuscript therefore provides an important contribution to a topic and area that definitely deserve extended consideration now and in the coming years, as nation states and religious communities are led to address the traumatic legacy of colonization.
To be more compelling, the manuscript needs to be subjected to a number of revisions/modifications. The first of these relates to the notion of “frozen trauma.” As it currently stands, the argument does not clearly distinguish frozen trauma from other kinds of trauma. It would be helpful to describe in greater detail the specific characteristics of frozen trauma and ways to perceive and heal it. When exactly does “regular” trauma become frozen? What needs to be done for traumatic experiences to move from a frozen state into a more explicit, workable state? In what ways does the processing/treatment of frozen trauma differ from that of other forms of traumatic experiences? The argument also associates frozen trauma with the overemphasis in contemporary culture on individual satisfaction and happiness, at the expense of relational interaction and the common good. Individual comfort is claimed to undergird proneness to exclude others on the basis of difference (taking multiple forms). What is not clear is the role trauma and frozen trauma in particular play in promoting or reinforcing this individualistic focus and proneness to exclusion. A more direct connection between these realities must be made and a sufficient rationale provided to make these claims convincing.
The argument makes sound use of Lévinas’ philosophy of the face, affirming the irreducible alterity of the human other and the intrinsically relational character of human identity and personhood. Human beings do not give themselves humanity, they receive it as a transcendent gift from God through one another. It would be important to also emphasize that the irreducible otherness of the human person does not entail the inexistence, but rather presupposes a common, shared humanity. The same, complete humanity is embodied in each individual in a unique and irreplaceable way. This unresolved and unresolvable tension between common humanity and singular individuality forms the necessary precondition to life-giving relationships between persons understood as infinite mysteries.
In section 4 (Accountability), the argument goes on to claim that formative relationship of other to other includes the recognition and healing of pain (and trauma). In lines 310-12, the following assertion is made: “Embodied recognition of the mutual others will therefore actively speak to repair the brokenness of the other, as it has also become my own brokenness.” While empathic knowledge and action undoubtedly form an essential component of the healthy processing of traumatic experiences, they cannot be accomplished through the denial of the alterity of the other. The argument needs to demonstrate that empathic recognition of the other’s trauma does not involve appropriation, for this would imply the infliction of further violence and trauma, perpetuating and reinforcing oppression rather than healing and overcoming it. Empathy must therefore be shown to include participation in someone else’s experience without this experience being assimilated to the experience and identity of the empathizing individual. The articulation of the relationship tying empathy to an ethics of responsible otherness demands further clarification and justification.
In section 5 (Justice) the following claim is made: “love also demands justice” (lines 400-01). Since at least Aristotle (his Nicomachean Ethics), the Western ethical tradition has affirmed that the (cardinal) virtue of justice acts as a necessary precondition for healthy human interactions, loving relationships and happiness. The author needs to clarify whether the argument reaffirms and/or adds anything to this longstanding tradition of thought. Lastly, in the concluding section, some implications of the argument for religious communities and the leadership they can exercise in enabling the processing of traumatic experiences are drawn. No explicit reference is made to the role religious communities and traditions played in the infliction of trauma, through colonization and oppression. It would be important (and arguably essential) to address the ways in which religious communities and traditions can become aware of their own traumatic and traumatizing history so as to become agents of healing and positive transformation for themselves before they attempt to heal others. Religious communities and traditions are not trauma and responsibility free; to be part of the solution, they must first acknowledge that they are part of the problem. Healing and transformation begin with oneself.
The manuscript needs significant linguistic revision. For suggested changes, please see the attached PDF file, in which I have added comments (using Adobe software annotating tools).

Author Response
Response to the comments and suggestions from the peer-reviewers
I want to start by thanking the peer reviewers for their very constructive comments and proposals regarding this contribution. This indeed valued and appreciated.
I have attended to all the comments indicated on the text.
In terms of the comments made in the reports I have tried my best to accommodate them in the attached version of the contribution.
In terms of the remark on the notion of “frozen trauma.” The reviewer requested the following: “As it currently stands, the argument does not clearly distinguish frozen trauma from other kinds of trauma. It would be helpful to describe in greater detail the specific characteristics of frozen trauma and ways to perceive and heal it. When exactly does “regular” trauma become frozen? What needs to be done for traumatic experiences to move from a frozen state into a more explicit, workable state? In what ways does the processing/treatment of frozen trauma differ from that of other forms of traumatic experiences?” I elaborated extensively on this notion in another publication. I have added a footnote in this regard, footnote 3.
In terms of the comment, “The argument also associates frozen trauma with the overemphasis in contemporary culture on individual satisfaction and happiness, at the expense of relational interaction and the common good. Individual comfort is claimed to undergird proneness to exclude others on the basis of difference (taking multiple forms). What is not clear is the role trauma and frozen trauma in particular play in promoting or reinforcing this individualistic focus and proneness to exclusion. A more direct connection between these realities must be made and a sufficient rationale provided to make these claims convincing” I have attempted to deal with this comment on page 4.
In terms of the comment: “The argument makes sound use of Lévinas’ philosophy of the face, affirming the irreducible alterity of the human other and the intrinsically relational character of human identity and personhood. Human beings do not give themselves humanity, they receive it as a transcendent gift from God through one another. It would be important to also emphasize that the irreducible otherness of the human person does not entail the inexistence, but rather presupposes a common, shared humanity. The same, complete humanity is embodied in each individual in a unique and irreplaceable way. This unresolved and unresolvable tension between common humanity and singular individuality forms the necessary precondition to life-giving relationships between persons understood as infinite mysteries.” I have addressed this on page 6.
In terms of the comment “In section 4 (Accountability), the argument goes on to claim that formative relationship of other to other includes the recognition and healing of pain (and trauma). In lines 310-12, the following assertion is made: “Embodied recognition of the mutual others will therefore actively speak to repair the brokenness of the other, as it has also become my own brokenness.” While empathic knowledge and action undoubtedly form an essential component of the healthy processing of traumatic experiences, they cannot be accomplished through the denial of the alterity of the other. The argument needs to demonstrate that empathic recognition of the other’s trauma does not involve appropriation, for this would imply the infliction of further violence and trauma, perpetuating and reinforcing oppression rather than healing and overcoming it. Empathy must therefore be shown to include participation in someone else’s experience without this experience being assimilated to the experience and identity of the empathizing individual. The articulation of the relationship tying empathy to an ethics of responsible otherness demands further clarification and justification.” The focus of my argument is not on empathic recognition but on embodied recognition. However I have attempted to deal with this on page 9.
In terms of the following comment “In section 5 (Justice) the following claim is made: “love also demands justice” (lines 400-01). Since at least Aristotle (his Nicomachean Ethics), the Western ethical tradition has affirmed that the (cardinal) virtue of justice acts as a necessary precondition for healthy human interactions, loving relationships and happiness. The author needs to clarify whether the argument reaffirms and/or adds anything to this longstanding tradition of thought.” I have addressed it on page 11.
In terms of this comment; “Lastly, in the concluding section, some implications of the argument for religious communities and the leadership they can exercise in enabling the processing of traumatic experiences are drawn. No explicit reference is made to the role religious communities and traditions played in the infliction of trauma, through colonization and oppression. It would be important (and arguably essential) to address the ways in which religious communities and traditions can become aware of their own traumatic and traumatizing history so as to become agents of healing and positive transformation for themselves before they attempt to heal others. Religious communities and traditions are not trauma and responsibility free; to be part of the solution, they must first acknowledge that they are part of the problem. Healing and transformation begin with oneself.” I have attempted to address this on page 12.
Reviewer 2 Report
I would not begin with Covid-19 in the way you did. I see what is intended but I am of the opinion that the way you do it does not help lead whether reader go where you want them to go. Your point is not to discuss COVID but to discuss something that the COVID context has revealed in South Africa. I would begin by describing the violence,paralysis, xenophobia and other behaviors that SA is experiencing. I would then go on to offer the disclaimer I hear you making that all this is not because of COVID context but that COVID has exposed them or that they have become acute during COVID. Your position is that all this is indicative of Frozen Generational Trauma.
What were the promises/prospects of the new South Africa which have not been realized that lead to the trauma? What were the promises/prospects of the TRC that have not been met that have led to this frozenness.
In the conclusion I expected some type of “prescription” but I do not hear one. The whole discussion is a “diagnosis” of this frozen trauma. I think you do a good job in that. The reader is left asking “So what? What do we do with this? What does this call for divine discomfort lead to?” The how question is not answered! Your last sentence in the abstract suggests you are going to make some suggestion about how the Religious, traditional and political spheres can ameliorate this frozenness. It’s not about an exhaustive prescription but an indication of how this “divine discomfort” can help deal with this inter generational trauma. For me. This would greatly improve your discussion.
Author Response
Response to the comments and suggestions from the peer-reviewers
I want to start by thanking the peer reviewers for their very constructive comments and proposals regarding this contribution. This indeed valued and appreciated.
I have attended to all the comments indicated on the text.
In terms of the comments made in the reports I have tried my best to accommodate them in the attached version of the contribution.
In terms of the comment: “In the conclusion I expected some type of “prescription” but I do not hear one. The whole discussion is a “diagnosis” of this frozen trauma. I think you do a good job in that. The reader is left asking “So what? What do we do with this? What does this call for divine discomfort lead to?” The how question is not answered! Your last sentence in the abstract suggests you are going to make some suggestion about how the Religious, traditional and political spheres can ameliorate this frozenness. It’s not about an exhaustive prescription but an indication of how this “divine discomfort” can help deal with this inter generational trauma. For me. This would greatly improve your discussion.” I particularly wanted to avoid engaging with the how as the how can dominate the argument. I tried to focus on what needs to happen before the how is developed. I also am of the opinion that the how needs to be developed from the bottom up.
I want to thank you again for comments and the recommendations. I sincerely hope I have addressed this accordingly.
Many greetings
Author